# Economic impact of cholera in households in rural southern Malawi: a prospective study

Amber Hsiao  ,[1] Enusa Ramani,[1,2] Hye-Jin Seo,[3] GiDeok Pak,[4] Dan Vuntade,[5] Maurice M'bang'ombe,[6] Bagrey Ngwira,[5] Wilm Quentin,[1] Florian Marks  ,[7,8] Vittal Mogasale  [2]

BN since deceased.

For numbered affiliations see end of article.

**Correspondence to**
Dr Vittal Mogasale;
vmogasale@gmail.com

## ABSTRACT

**Introduction** Cholera remains a significant contributor to diarrhoeal illness, especially in sub-Saharan Africa. Few studies have estimated the cost of illness (COI) of cholera in Malawi, a cholera-endemic country. The present study estimated the COI of cholera in Nsanje, southern Malawi, as part of the Cholera Surveillance in Malawi (CSIMA) programme following a mass cholera vaccination campaign in 2015.

**Methods** Patients ≥12 months of age who were recruited as part of CSIMA were invited to participate in the COI survey. The COI tool captured household components of economic burden, including direct medical and non-medical costs, and indirect lost productivity costs.

**Results** Between April 2016 and March 2020, 40 cholera cases were enrolled in the study, all of whom participated in the COI survey. Only two patients had any direct medical costs and five patients reported lost wages due to illness. The COI per patient was US$14.34 (in 2020), more than half of which was from direct non-medical costs from food, water, and transportation to the health centre.

**Conclusion** For the majority of Malawians who struggle to subsist on less than US$2 a day, the COI of cholera represents a significant cost burden to families. While cholera treatment is provided for free in government-run health centres, additional investments in cholera control and prevention at the community level and financial support beyond direct medical costs may be necessary to alleviate the economic burden of cholera on households in southern Malawi.

## INTRODUCTION

Cholera is an infection of the intestine caused by the bacterium *Vibrio cholerae* that leads to acute, rapidly dehydrating diarrhoeal disease. Annually, an estimated 2.86 million cases of cholera occur in cholera-endemic countries that result in 95 000 deaths.[1] A recent study estimated that cholera is especially pervasive throughout sub-Saharan Africa (sSA) where the human and economic burden of cholera is greatest, with more than 200 million people living in areas where cholera is present. Of these 200 million, 87 million live in areas with

high incidence, defined as >1 case per 1000 people per year.[2]

## STRENGTHS AND LIMITATIONS OF THIS STUDY

⇒ We recruited patients who were cholera-confirmed at the time they first sought care and interviewed them regarding costs incurred due to illness within 7 days following their initial health facility visit, which limited recall bias.

⇒ The sample size was relatively small, which limits the generalisability of the findings.

⇒ The cost of illness estimate is likely to be an underestimate as it only includes out-of-pocket expenditures of patients who sought care.

While improvements to water, sanitation and hygiene (WASH) are the best long-term solution for preventing more diarrhoeal diseases, these measures remain unavailable to large proportions of people in sSA, especially among those living in flood-prone rural areas. The availability of low-cost inactivated oral cholera vaccines (OCVs) since 2011 has significantly changed the landscape and provided a cost-effective short-to-middle-term approach for cholera prevention and control.[3] A number of mass vaccination campaigns—many of which have taken place in sSA—have demonstrated that it can be successfully administered in a variety of settings, including high-risk settings to prevent an impending outbreak, or to quickly halt the transmission of an ongoing outbreak.[4 5]

In Malawi specifically, cholera has been reported since the 1970s and is endemic to the country.[6–8] The extent of the cholera burden in the country has not been well documented due to the lack of sustainable diagnostic and surveillance capacity.

One of the worst cholera outbreaks to date occurred in January 2015 following the nation's worst floods in history, which affected more than 1.1 million people in the Southern Region and displaced at least 230 000 people

BMJ

from their homes.[9] Within a month of the flood, cholera cases were being reported in Nsanje and Chikwawa, two of the worst-hit districts in southern Malawi. The flooding led to a multistakeholder response to coordinate a reactive OCV mass vaccination campaign in the region.[10] Due to the lack of diagnostic capacity, cholera cases were being diagnosed based on clinical presentation as per the WHO and national guidelines.[11]

The OCV mass vaccination campaign was then implemented primarily in Nsanje, which was identified as having the highest burden of cholera based on prior years' data provided by the Malawian Ministry of Health (MOH). From 30 March 2015 to 23 April 2015, the International Vaccine Institute (IVI), MOH, WHO and John Snow, Inc coordinated a reactive mass vaccination campaign that targeted 160 000 people (320 000 total doses of the OCV Shanchol (Shantha Biotechnics Pvt, Ltd., India) over 1 year of age. Those living in high-risk areas of Nsanje—namely, those living in internally displaced camps and surrounding villages—were the primary targets.[10]

In the course of a 4-year effort to strengthen cholera surveillance capacity and infrastructure in Nsanje following the mass vaccination campaign, we investigated the cost of illness (COI) of cholera for patients who sought care. Only one study to date has reported on the COI from cholera in southeastern Malawi (Lake Chilwa in the districts of Machinga and Zomba).[12] The objective of this study is to provide additional field data on the economic burden of cholera in Malawi from a household perspective.

## METHODS

As part of the Cholera Surveillance in Malawi (CSIMA) programme, cholera surveillance was conducted across 22 Nsanje (population >292 000) and 18 Chikwawa (population >533 000) healthcare facilities from 5 April 2016 to 31 March 2020. IVI staff trained health surveillance assistants (HSAs) and health workers at all Nsanje and Chikwawa facilities to identify patients who sought care for acute watery diarrhoea. HSAs are community health facilitators who live in the area served by the local healthcare facility who are involved in activities such as WASH and general health promotion.

Patients who met the inclusion criteria (who were eligible to receive OCV based being at least 12 months of age at the start of the vaccination campaign, and presented to an Nsanje health facility with acute watery diarrhoea indicative of cholera as defined by the WHO[11]) were invited to participate in the diarrhoeal surveillance study. Participants who provided informed written consent provided a stool specimen to perform a rapid diagnostic test (RDT) for cholera (Crystal VC RDT; Arkray Healthcare Pvt, India; previously Span Diagnostics, Surat, India) and/or a stool sample for culture confirmation at the national reference laboratory in Lilongwe. The use of RDTs has been accepted for use by the Malawi MOH, and there is documented experience with this test kit in other

settings.[13 14] While the use of the RDT is not intended to be a substitute for stool culture, the use of the kit in this setting ensured that one would not miss any potential cholera cases during the early surveillance period while logistics for stool transportation and culture were being set-up.

Between 6 April 2016 and 12 December 2017, culture-confirmed and RDT-positive cholera cases were recruited for the overall study, as well as the COI survey component. However, due to low recruitment numbers of RDT-confirmed and/or culture-confirmed cholera cases, in part due to limited availability of RDT kits and stool collection logistical challenges, the inclusion criteria were expanded from 12 December 2017, through 31 March 2020, to include clinically diagnosed cholera cases in an outbreak situation to avoid missing potential cases.

While cholera vaccines were administered to residents in both Nsanje and Chikwawa, the majority of doses were administered to residents of Nsanje. Thus, only patients recruited from Nsanje health centres were invited for the COI portion of the study. The COI from a household perspective was assessed for patients who provided written informed consent (regardless of vaccination status). The COI tool (online supplemental material) was developed based on a tool that IVI had previously used for other COI studies, with further refinements using the guidelines for cost collection in cholera studies provided by the WHO.[15 16] The survey tool captured household components of economic burden, divided further into direct medical costs (DMCs), direct non-medical costs (DNMCs) and indirect costs (ICs) (as done in the previous studies[17]), with the payer being the patients themselves or caregivers/helpers of the patient (eg, relatives). Experienced interviewers were trained on how to administer the survey and all survey responses were validated by the University of Malawi Polytechnic and MOH collaborators to ensure they were appropriate for the local context.

A series of a maximum of three interviews were planned for the COI assessment, with day 0 (ie, day of diagnosis) being the day of the patient's health facility visit: at day 0–7, day 13–15 (beyond the mean duration of an episode of cholera illness) and day 20–22. To minimise recall bias, the first interview took place as soon as cholera was confirmed either by a test or a clinician, with a maximum of 7 days following the first healthcare facility visit. If a patient preferred to be interviewed while at the healthcare facility, the trained interviewer conducted the first COI interview at the time of the initial outpatient or inpatient visit. Patients who were not feeling well enough were contacted by an HSA to schedule a future interview time. Trained interviewers would then visit the patient's home at the scheduled time to conduct the first interview. At the end of each COI assessment, patients were asked whether they felt they had fully recovered from cholera. If the patient reported still being ill, the next interview was scheduled; otherwise, no additional interviews were conducted.

At each of the interviews, patients (or parent/guardian) were asked how much money they spent during each of their visits to a healthcare facility or other treatment locations (eg, a traditional healer) since becoming ill, including any visits that may have occurred prior to the study enrolment date. DMCs included out-of-pocket payments for services, prescription drugs and any other health facility fees. DNMCs included transportation and food expenses incurred by the patient and/or caregivers who went to the health facility with the patient.

Finally, IC accounted for the patients' and/or caregivers' productivity losses incurred due to lost wages. We calculated productivity losses using three scenarios based on (1) self-reported wage losses, (2) minimum wage per day for all workers between 14 and 50 years of age (the legal minimum and old-age pension eligibility age[18]) who did not report wage losses and (3) minimum wage per day for all patients >5 years of age who did not report wage losses, given evidence from Africa that children are often involved in economic activities that add to a family's income.[19]

In all scenarios, if patients and/or caregivers reported wage losses, these were calculated based on self-reported daily lost productivity (half or full days of work lost due to taking care of the ill patient) and daily wages. For patients and caregivers who did not report wage losses (ie, scenarios 2 and 3), we applied the minimum wage of 1346.16 Malawian kwacha (MWK) per day (US$1.83/day as of 1 January 2020) to account for the monetary value of productivity losses.[20] Productivity losses were not calculated for caregivers who indicated that they did not cut back on their usual activities had they not been caring for the patient. They were also not calculated for caregivers who reported no income lost (eg, a caregiver who is dedicated to unpaid housework).

Costs were collected in MWK, inflated to 2020 MWK using World Bank inflation data,[21] and converted to 2020 US$ (736.5803 MWK=US$1 on 31 March 2020).[22] All analyses were conducted in Stata/IC 13.1 (College Station, Texas, USA) and Microsoft Excel.

### Patient and public involvement

Patients including their caretakers were involved in this study as participants in the COI interviews. Costs related to cholera illness were collected from the participants after obtaining informed consent. The public was not involved in this study. Patients and the public were not involved in study development, defining the research question and study design, recruitment and the conduct of the study. Early findings of the study were disseminated through a workshop attended by country policymakers, health managers and health facility staff.

### RESULTS

During our surveillance period, 2201 individuals met the diarrhoeal illness criteria for stool collection and testing for cholera in Nsanje. Of these patients, 40 (1.8%) were identified as cholera cases and were recruited for the surveillance study. On the day of their health facility visit, all 40 patients also consented for COI interview. Table 1 summarises selected demographic characteristics, as well as OCV statuses, of all cases.

Of the 40 cases, 31 (78%) were seen in an outpatient setting. Duration of illness was 5 days (min–max: 2–17 days) in the outpatient setting and 8 days (min–max: 2–19 days) in the inpatient setting. Most patients (n=34) had fully recovered by the time the first interview occurred (scheduled within 7 days of recruitment day 0); 6 patients had a second interview (at day 13–15) and none required a third interview. A higher proportion of inpatient cases were males, whereas outpatient cases were majority female. Of the 40 cases, 4 were identified clinically, 9 using RDT and 27 using culture confirmation. A larger proportion of children were seen in the inpatient setting, compared with outpatient.

The direct and IC of cholera per patient was US$14.34 when averaged across all patients, with the largest proportion of costs (57.9%) being from DNMCs (food, water and transportation) (table 2). DMCs for drugs and consumables were zero for nearly all patients; only two patients seen in the inpatient setting had any drug costs of US$11.17 (8229 MWK) and US$1.19 (875 MWK) each, both of whom were seen at a private Christian Health Association of Malawi (CHAM) facility; these two patient costs increased the overall average across the cohort to US$0.31.

Patient and caregiver food costs were US$0.95 and US$1.61 on average, respectively. Roundtrip transportation costs were relatively lower for patients at US$1.84, compared with caregiver transportation costs at US$3.91. Transportation time on average per patient was 112 min roundtrip (min, max: 10–360).

We also performed various scenario analyses for ICs (table 3), where costs were only averaged across patients and caregivers with non-zero productivity losses in each scenario. Eight patients were excluded from wage loss calculations due to missing age or birthdate. Thirty-four patients had at least one caregiver provide care during their illness, with a total of 53 caregivers across all patients. Caregivers only reported whether they were children or adults (ie, exact ages or birthdates were not ascertained); all caregivers reported being adults who we assumed were of working age.

In scenario 1, productivity losses (lost wages) due to being sick for the duration of illness were US$10.60 on average (n=5) for patients who reported wage losses, though one patient reported receiving paid sick leave for this time (US$10.89), which would reduce the average to US$10.53 (n=4). Only 15 caregivers (who cared for eight patients) reported US$6.36 on average in lost wages due to missing work. In scenario 2, where the minimum wage was assumed for all patients (n=11) and caregivers (n=45) of working age, productivity losses were US$8.83 for patients and US$3.94 for caregivers on average. In scenario 3, where the minimum wage was assumed for all

**Table 1** Demographic and medical characteristics of cholera cases in Nsanje District, Malawi, 2016–2020

|  | Outpatient n (%) | Inpatient n (%) | Total* n (%) |
|---|---|---|---|
| Total | 31 (78) | 9 (22) | 40 (100) |
| **Sex** | | | |
| Male | 9 (29) | 6 (67) | 15 (38) |
| Female | 17 (55) | 3 (33) | 20 (50) |
| Unknown/missing | 5 (16) | 0 (0) | 5 (13) |
| **Age** | | | |
| <5 years | 10 (32) | 4 (44) | 14 (35) |
| ≥5 years to <18 years | 2 (6) | 1 (11) | 3 (8) |
| ≥18 years to <35 years | 9 (29) | 0 (0) | 9 (23) |
| ≥35 years | 3 (10) | 3 (33) | 6 (15) |
| Unknown/missing | 7 (23) | 1 (11) | 8 (20) |
| **Level of education** | | | |
| Never went to school | 13 (42) | 3 (33) | 16 (40) |
| Preschool | 2 (6) | 0 (0) | 2 (5) |
| Completed primary | 10 (32) | 3 (33) | 13 (33) |
| Completed secondary | 2 (6) | 0 (0) | 2 (5) |
| Unknown/missing | 4 (13) | 3 (33) | 7 (18) |
| **Employment status** | | | |
| Works for a wage | 5 (16) | 0 (0) | 5 (13) |
| Does not work for a wage | 26 (84) | 9 (100) | 35 (88) |
| Average inpatient stay, days (min–max) | – | 2 (1–4) | – |
| Average number of symptomatic days prior to seeking medical care (min–max) | 1.4 (0–4) | 2.8 (1–8) | 1.7 (0–8) |
| Average number of days sick with cholera | 5 (2–17) | 8 (2–19) | 6 (2–19) |
| **Method of transportation for patients to health centre** | | | |
| Walked | 9 (29) | 1 (11) | 10 (25) |
| Biked | 12 (39) | 5 (56) | 17 (43) |
| Motorcycle | 3 (10) | 1 (11) | 4 (10) |
| Minibus | 5 (16) | 1 (11) | 6 (15) |
| Private car | 1 (3) | 0 (0) | 1 (3) |
| Hired canoe | 1 (3) | 0 (0) | 1 (3) |
| Ambulance (from MOH) | 0 (0) | 1 (11) | 1 (3) |
| **Vaccination status (OCV in 2015)** | | | |
| Not vaccinated | 16 (52) | 5 (56) | 21 (53) |
| One dose | 5 (16) | 1 (11) | 6 (15) |
| Two doses | 6 (19) | 0 (0) | 6 (15) |
| Unknown/missing | 4 (13) | 3 (33) | 7 (18) |
| **Cholera confirmation** | | | |

Continued

**Table 1** Continued

|  | Outpatient n (%) | Inpatient n (%) | Total* n (%) |
|---|---|---|---|
| Rapid diagnostic test | 5 (56) | 4 (44) | 9 (22) |
| Lab culture confirmation | 25 (93) | 2 (7) | 27 (68) |
| Clinically identified | 1 (25) | 3 (75) | 4 (10) |

*Sum of percentages may exceed 100 due to rounding.
MOH, Ministry of Health; OCV, oral cholera vaccine.

patients >5 years of age (n=19), productivity losses were US$9.78 for patients on average. In this scenario analysis, there were two patients aged 5–14 years (US$14.16 on average) and five patients aged ≥60 years (US$8.59 on average). Caregiver costs in scenarios 2 and 3 were the same (US$3.94) since all caregivers were adults.

Half of all patients (n=20), and two-thirds of inpatients (n=6), reported having to borrow money from someone to pay for healthcare, food or transportation as a result of having cholera (table 4), but most did not have to sell personal belongings to pay for care. Experiencing at least some food insecurity in the past year (defined as going without food at least once in the past 3 months) was an issue for a large proportion of patients (n=28).

## DISCUSSION

Our study estimated patient and caregiver costs associated with seeking healthcare for the treatment of severe diarrhoeal illness diagnosed as cholera in Nsanje. We had few diarrhoeal cases overall. The cost of cholera illness per patient was US$14.34, with the largest proportion of costs being from DNMCs (food, water and transportation) (US$8.30), followed by indirect productivity losses (US$5.73) in our primary analysis, where costs were averaged across all patients in our cohort.

The majority of our patients were not admitted during their visit to the health centre. Patients who sought care in the outpatient setting were symptomatic an average of 1.4 days prior to seeking care. Patients who were admitted were sick for an average of 2.8 days prior to seeking care and had an inpatient stay of 2 days. Our patient population was slightly skewed towards females and of those whose age was reported, most patients were adults aged 18 and older. The majority of patients were not vaccinated during the OCV campaign in 2015.

Overall, only two patients with cholera had any DMCs; one was seen in the outpatient setting (875 MWK or US$1.19) and the other was admitted to a private CHAM facility (8229 MWK or US$11.17). This finding was expected, as cholera treatment is provided free of charge in Malawi. On average, patients and their caregivers spent US$8.61 in direct costs (US$0.31 in medical costs and US$8.30 in non-medical costs, including those with zero costs), almost all of which was due to food and/or water and transportation costs. In our scenario analysis

**Table 2** Mean direct and indirect costs (MWK and USD 2020) for cholera cases in Nsanje District, Malawi, 2016–2020* (averaged across all 40 patients)

| | Inpatient (n=9) | | Outpatient (n=31) | | Total costs (n=40) | | |
|---|---|---|---|---|---|---|---|
| | 2020 MWK (min, max) | 2020 USD (min, max) | 2020 MWK (min, max) | 2020 USD (min, max) | 2020 MWK | 2020 USD | % of total |
| Direct, medical costs | 97 | 0.13 | 265 | 0.36 | 228 | 0.31 | 2.2 |
| Drugs and consumables† | 97 (0–875) | 0.13 (0–1.19) | 265 (0–8229) | 0.36 (0–11.17) | 228 | 0.31 | 2.6 |
| Direct, non-medical costs | 9703 | 13.17 | 5073 | 6.89 | 6115 | 8.30 | 57.9 |
| Patient food and/or water | 505 (0–1353) | 0.69 (0–1.84) | 753 (0–6148) | 1.02 (0–8.35) | 697 | 0.95 | 6.6 |
| Patient transportation (roundtrip) | 2007 (0–8750) | 2.73 (0–11.88) | 1161 (0–9836) | 1.58 (0–13.35) | 1352 | 1.84 | 12.8 |
| Caregiver food and/or water | 1332 (0–2734) | 1.81 (0–3.71) | 1143 (0–12 686) | 1.55 (0–17.22) | 1186 | 1.61 | 11.2 |
| Caregiver transportation (roundtrip) | 5858 (0–13 525) | 7.95 (0–18.36) | 2016 (0–8346) | 2.74 (0–11.33) | 2880 | 3.91 | 27.3 |
| Indirect (productivity losses)‡ | 489 | 0.66 | 5304 | 7.20 | 4221 | 5.73 | 40.0 |
| Patient | 0 (0–0) | 0 (0–0) | 1259 (0–12 519) | 1.71 (0–17.00) | 976 | 1.32 | 9.2 |
| Caregivers | 489 (0–2213) | 0.66 (0–3.00) | 4045 (0–70 526) | 5.49 (0–95.75) | 3245 | 4.41 | 30.7 |
| Total mean costs§ | 10 289 | 13.97 | 10 643 | 14.45 | 10 563 | 14.34 | 100.0 |

*All costs by category are averaged over the total number of patients in each category (ie, patients or caregivers with zero costs are not excluded in the average).
†Only two patients (one inpatient, one outpatient) had any DMCs, but costs are averaged over the total number of patients.
‡Productivity losses are averaged over the total number of patients in each category (not averaged over the total number of caregivers); six outpatients and two inpatients had caregivers who reported wage losses, but wages are averaged over the total number of outpatients (n=31) and inpatients (n=9).
§Costs are first summed across categories for each patient and averaged across total patients.
DMCs, direct medical costs; MWK, Malawian kwacha.

when only those with self-reported wages are considered, only five patients with cholera (all outpatient cases) reported productivity losses of US$8.83 on average, and 15 caregivers of patients reported wage losses, averaging US$3.93. In our scenario analysis, where we applied the minimum wage to all patients and caregivers who

**Table 3** Scenario analyses of mean indirect costs for cholera cases with reported wage losses and assuming minimum wage in Nsanje District, Malawi, 2016–2020

| | Scenario 1: among those with self-reported wages only* Patients: n=5 Caregivers: n=15 | | % of total indirect costs | Scenario 2: self-reported + minimum wage for patients aged 15–50 years Patients: n=11 Caregivers: n=45 | | % of total indirect costs | Scenario 3: self-reported + minimum wage for patients aged >5 years Patients: n=19 Caregivers: n=45 | | % of total indirect costs |
|---|---|---|---|---|---|---|---|---|---|
| | 2020 MWK (min, max) | 2020 USD (min, max) | | MWK (min, max) | 2020 USD (min, max) | | MWK (min, max) | 2020 USD (min, max) | |
| **Indirect, productivity losses** | | | | | | | | | |
| Patients* | 7807 (5008–12 519) | 10.60 (6.80–17.00) | 63 | 6506 (2692–12 519) | 8.83 (3.66–17.00) | 69 | 7161 (2019–16 154) | 9.78 (2.74–21.93) | 71 |
| Caregivers | 8558 (823–25 039) | 6.36 (1.12–33.99) | 38 | 5306 (823–25 039) | 3.94 (1.12–33.99) | 31 | 5306 (823–25 039) | 3.94 (1.12–33.99) | 29 |
| Total indirect costs | 16 365 | 16.96 | | 11 812 | 12.77 | | 12 467 | 13.72 | |

*All 5 patients who worked for a wage were outpatient.
MWK, Malawian kwacha.

**Table 4** Financial and food insecurity of cholera cases in Nsanje District, Malawi, 2016–2020

| | Outpatient (n=31) n (%) | Inpatient (n=9) n (%) | Total (n=40) n (%) |
|---|---|---|---|
| Borrowing money to pay for healthcare services | | | |
| Did not borrow money | 17 (55) | 3 (33) | 20 (50) |
| Borrowed money from a family member | 6 (19) | 0 (0) | 6 (15) |
| Borrowed money from a friend | 4 (13) | 5 (56) | 9 (23) |
| Borrowed money from a cooperative | 0 (0) | 1 (11) | 1 (3) |
| Borrowed money from a neighbour | 1 (3) | 0 (0) | 1 (3) |
| Borrowed money from the village chief | 1 (3) | 0 (0) | 1 (3) |
| Borrowed money from boss/employer | 1 (3) | 0 (0) | 1 (3) |
| Borrowed money from church | 1 (3) | 0 (0) | 1 (3) |
| Sold items to pay for healthcare services | | | |
| Did not sell anything | 29 (94) | 4 (44) | 33 (83) |
| Rented out rice and maize plots | 0 (0) | 4 (44) | 4 (10) |
| Sold radio | 1 (3) | 0 (0) | 1 (3) |
| Sold livestock | 1 (3) | 1 (11) | 2 (5) |
| Frequency of household going without food in the past year | | | |
| Never | 4 (13) | 1 (11) | 5 (13) |
| Once every 3 months | 7 (23) | 1 (11) | 8 (20) |
| Once per month | 3 (10) | 2 (22) | 5 (13) |
| Twice per month | 2 (6) | 1 (11) | 3 (8) |
| Once per week | 8 (26) | 1 (11) | 9 (23) |
| Almost every day | 3 (10) | 0 (0) | 3 (8) |
| Unknown/missing | 4 (13) | 3 (33) | 7 (18) |

theoretically could be contributing to economic activities (ie, household work), indirect productivity losses were as high as US$9.78 per patient and US$3.94 per caregiver. This latter scenario may overestimate the number of individuals who experience productivity losses and the actual economic impact, though the average productivity losses (US$13.72) were lower than in the self-reported wage scenario ($16.96).

The overall costs found in our study are substantially less than one study previously reported in rural Malawi.[12] Ilboudo et al reported household costs of US$65.60 in 2016 (US$106.16 in 2020). They reported a much larger proportion of hospitalised cases: their study sampled 100 cases (which estimated 9% of the overall caseload from the cholera outbreak), 90 of whom were hospitalised, whereas we only had 7 hospitalised cases (17.5% of total cases). Transportation costs in our study were also lower for both patients and caregivers compared with their study, which may be attributable to differences in transportation methods and travel distances to health centres.

The largest differences in costs between our study and Ilboudo et al were in food and lost productivity costs, likely also driven by the fact that their study had more hospitalised cases that required more expenses over several days. Their reported ICs of patient's productivity were a larger portion of overall costs (35.1% compared with our 9.2%), which may be attributed to their study cohort having longer inpatient stays. The caregiver lost productivity costs as a proportion of overall costs were lower in their study (23.5% compared with our 30.7%). Differences in productivity costs may be attributed to several reasons, such as differences in wages in Nsanje versus Machinga and Zomba districts, unemployment rates and other socioeconomic factors. The differences may also be attributed to the severity of illness, potentially attenuated by vaccination. Ilboudo et al assessed costs of illness from patients who had cholera in the ±1 week prior to and after the OCV campaign that was underway (which followed the cholera outbreak). Thus, the proximity in timing to the outbreak may have meant that few individuals had any immunity to cholera and may have experienced more severe symptoms, whereas in our population, OCV coverage and immunity was likely higher. Given the relatively small sample size in both studies, however, more data are needed from the region to understand why these costs differed more than other cost components.

We also assessed financial and food insecurity for cases. Half of our cases had to borrow money from someone to pay for costs associated with their cholera illness and treatment, which is similar to the proportion reported by Ilboudo et al. A large proportion of our cases also faced food insecurity at some point in the prior year.

Our study had several limitations. One limitation was the late start of cholera surveillance (April 2016) relative to the cholera outbreak (approximately February 2015), in response to which a vaccination campaign was conducted (March/April 2015). It is possible that some postvaccination campaign cholera cases may have occurred in the year prior to the start of cholera surveillance, particularly during the November 2015 to April 2016 rainy season; these cases would not have been included in this study. The study was also not able to collect COI data from Chikwawa residents and infants <12 months, which may have provided different cost estimates. Given the small number of cholera cases, even after expanding the case criteria to clinically diagnosed cases (of which there were only 4), it may not be representative of the general Nsanje population, particularly those who were sick and did not seek care. It is also possible that patients may have shown up for care but were undiagnosed or misdiagnosed for reasons such as diagnostic sensitivity. While we were unable to assess any patterns or variation in the use of the three testing methods (RDT, culture or clinically identified) by the health centre, it is possible that our recruited patient cohort is biased towards health centres that were better equipped to test, recruit and treat patients. For example, many of the more remote health centres in Nsanje lacked toilets or latrines for proper stool collection and have limited access to electricity to ensure the viability of stool samples, which may have limited the sensitivity of the stool culture result. Apart from the logistical challenges of recruitment, the area also has a high number of migrants from Mozambique; migrants may be less likely to seek care due to several reasons, including language barriers, discrimination due to cultural differences, and fear of deportation.[23]

Health facility costs were also not included in our costing, which may be a significant portion of the total costs as most patients reported US$0 in DMCs. A recent study in rural Malawi estimated that the average treatment cost to a health facility for a hospitalised cholera case was US$59.70 in 2016 (US$96.61 in 2020).[12] A separate study on cholera treatment costs at the facility level at this study site has estimated an economic cost of US$41.86 in 2018 (US$50.79 in 2020) and a financial cost of US$25.43 in 2018 (US$30.85 in 2020).[24] Finally, there was the potential for recall bias, especially for patients who may have had costs prior to the initial visit to the health centre from which they were recruited for the study. For costs incurred from the day of enrolment onward, we attempted to minimise the potential for recall bias by following up first within a week of the initial visit.

Despite the limitations, our study provides one of the few COI studies of cholera illness in sSA from the household perspective. More studies are needed to more fully evaluate the cholera costs of illness in Malawi. Currently, spending on health accounts for less than 10% of country revenues—below the 15% target set by the Abuja Declaration to which Malawi is a signatory.[25] Healthcare financing in Malawi is also unpredictable and unsustainable given the country's reliance on development partners' contributions that account for nearly two-thirds of total health expenditure.[26] Our study supports the need for further spending in the health sector; for example, building and staffing additional health centres that reduce the amount of time patients have to travel could reduce the cost burden on patients.

While the study was not designed to assess vaccine effectiveness, our surveillance efforts may also be a positive indication that the cholera vaccine may be associated with a reduction in the burden of cholera in the middle-term, as most of our cases were unvaccinated (while approximately half of the Nsanje population received the cholera vaccine). Since this study was conducted, in recognition of the high incidence of cholera and the need for additional tools to address the burden, the government of Malawi integrated OCVs into its national cholera control plan in 2017,[27] and there have been several OCV campaigns implemented since.[10 28] Alternatively, given that cholera often occurs cyclically, it is possible that the time period during which surveillance occurred was a low or no cholera transmission interval. Ultimately, investing more heavily in WASH infrastructure would be the most effective, though more costly, solution in the long-run to avert cholera.

In a country where the gross domestic product per capita per day is less than US$2 (US$625 annually in 2020), a family who has to spend US$14 on average for cholera illness will very likely have to make financial sacrifices to access treatment. Even though the DMCs associated with cholera care are low, the patient (who is willing and able to go to the health centre) must bear a high burden of DNMCs and indirect lost productivity costs. For those who do not have the means of even going to a health facility to seek care, the consequences are likely even more severe; for example, the death of a family member whose wages supported the family has significant long-term impacts beyond a single cholera episode.

In conclusion, we find that households face a significant cost burden when one falls sick with cholera. These costs are significantly high to warrant additional investments in the health system to avert future cases of cholera.

**Author affiliations**
[1]Department of Health Care Management, Technische Universität Berlin, Berlin, Germany
[2]Policy and Economic Research, International Vaccine Institute, Seoul, The Republic of Korea
[3]Epidemiology, Public Health, Impact, International Vaccine Institute, Seoul, The Republic of Korea
[4]Biostatistics & Data Management, International Vaccine Institute, Seoul, The Republic of Korea
[5]Department of Environmental Health, University of Malawi the Polytechnic, Blantyre, Malawi
[6]Ministry of Health Malawi, Lilongwe, Malawi
[7]Epidemiology, Public Health, Impact, International Vaccine Institute, Gwanak-gu, The Republic of Korea
[8]Cambridge School of Clinical Medicine, Cambridge Biomedical Campus, Cambridge Institute of Therapeutic Immunology and Infectious Disease, Cambridge, UK

**Contributors** AH and VM conceptualised the study, led the protocol development, field training and survey implementation in collaboration with H-JS, GDP, DV, MM, BN, FM and ER. Data collection was supervised by DV, MM and BN, and monitored by GDP, AH, VM, ER and H-JS. Data analysis and interpretation were done by AH supported by VM and WQ. The manuscript was written by AH, supported by VM and WQ. All authors have reviewed, edited and approved the final manuscript. AH and VM are the guarantors and take full responsibility for the finished work and/or the conduct of the study, had access to the data, and controlled the decision to publish.

**Funding** This research was conducted by International Vaccine Institute and funded by Bill & Melinda Gates Foundation (OPP38590) and the UK Foreign, Commonwealth and Development Office and Wellcome (214662/Z/18/Z). The International Vaccine Institute acknowledges its donors including the Republic of Korea, Swedish International Development Cooperation Agency and Republic of India.

**Competing interests** None declared.

**Patient and public involvement** Patients and/or the public were involved in the design, or conduct, or reporting or dissemination plans of this research. Refer to the Methods section for further details.

**Patient consent for publication** Consent obtained directly from patient(s).

**Ethics approval** This study was approved by the National Health Sciences Research Committee (NHSRC) of the Ministry of Health in Malawi (protocol #16/2/1547; approval #1547) on 29 March 2016. The International Vaccine Institute's (IVI) Institutional Review Board (IRB) approved the study on 6 April 2016. All participants (or a parent/guardian) provided written informed consent prior to interviews. For patients aged <12 years, consent was provided by the parent/guardian, and in these cases, the patient/caregiver provided the interview responses. For patients aged 12–17 years, ascent was provided by both the parent/guardian and child. Adults aged ≥18 years provided consent directly. The IVI IRB and the NHSRC approved the modification to expand to inclusion of clinically diagnosed cases on 31 May 2017 and 31 July 2017, respectively. Training of all health surveillance assistants on the modification occurred on 12 and 13 December 2017, and the data collection continued until 31 March 2020.

**Provenance and peer review** Not commissioned; externally peer reviewed.

**Data availability statement** All data relevant to the study are included in the article or uploaded as supplementary information.

**ORCID iDs**
Amber Hsiao http://orcid.org/0000-0003-2295-0886
Florian Marks http://orcid.org/0000-0002-6043-7170
Vittal Mogasale http://orcid.org/0000-0003-0596-8072

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
