## [Reviewer comments · BMJ Open]

ARTICLE DETAILS

TITLE (PROVISIONAL)	The economic impact of cholera in households in rural southern Malawi: a prospective study
AUTHORS	Hsiao, Amber; Ramani, Enusa; Seo, Hye-jin; Pak, GiDeok; Vuntade, Dan; M'bang'ombe, Maurice; Ngwira, Bagrey; Quentin, Wilm; Marks, Florian; Mogasale, Vittal

VERSION 1 – REVIEW

REVIEWER	Awalime, Dziedzom The Aurum Institute, International
REVIEW RETURNED	11-Aug-2021

GENERAL COMMENTS	The manuscript asserts that Malawi experienced its worst outbreak of cholera in 2015, this should reflect in larger case numbers being sampled, however, only 40 cases were sampled. A focus on cost-saving to households due to the mass vaccination programme undertaken would have been a good approach for this study. Since it could be attributed to the relatively lower number of cases reported. The data collection process was good and has minimized recall bias. The interval interview process helps for appropriate data to be captured. However, it is not clear if patients that were interviewed at the time of initial outpatient or inpatient visit, were followed-up since this has the potential of missing out on other costs that may occur after this initial phase of care? The interview for costs should be close-ended and must list possible cost drivers. Asking generally on how much money was spent has the tendency for the introduction of unrelated costs or missing out on related costs. Productivity losses are normally associated with 'time' paid jobs or informal sectors. Formal sector are normally entitled to sick days leave hence the individual does not bare productivity losses. The employment status of the sample shows that there were 35 out of 40 patients who work for wages. This does not come out clearly to signify if this cohort experienced productivity losses. Caregiver productivity losses are however captured. Though generally, children support the economic activity of parents/caregivers, this is usually done in the context of 'giving a hand' to support the family business. These children do not earn wages and the consideration of this as economic activity has encountered some criticisms especially with child rights advocates. In addition, these kinds of 'hand giving' are usually not contract time-bound. More than 50% (27) of the sample incurred no transportation cost which is usually an important cost driver. The conclusions of this study are issues widely known. Though the context of Malawi is being presented in this study, the small
--

	sample size makes it challenging to use the results from the study as the generalization of the Malawian situation, which can be totally different from what this study has postulated.
--	---

REVIEWER	Mukhopadhyay, Debraj Delhi Pharmaceutical Sciences and Research University
REVIEW RETURNED	19-Aug-2021

GENERAL COMMENTS	Fair enough to write this.
----------------------------

REVIEWER	Mejia, Nelly Centers for Disease Control and Prevention
REVIEW RETURNED	29-Nov-2021

GENERAL COMMENTS	This manuscript is relevant to understand the economic burden that patients experience during a case of cholera in Malawi. I think the study is well done, although may not be replicable because the volume of cholera cases depend on many external variables. My main comment is that the authors should clarify some parts of the methods and results to make the study more useful. Cost-of-illness of cholera in rural southern Malawi from the household perspective The authors explore an important topic: the costs of a cholera case from the patient/household perspective in a low-income country. Most studies about cost of illness for different diseases focus on the costs for the provider instead of the patient. The manuscript is well written, and the findings provide a good idea of how burdensome a case of cholera can be for a household in Malawi, even if the direct medical costs are free (in most cases). However, some parts of the methods are unclear and it would be good if the authors add some clarifications to be sure their methods are consistent and are comparing apples to apples. Specific comments below: Abstract:  • Please clarify in what year are the values provided in dollars (line 38). Is it 2021 USD? Introduction:  • Line 63: "Annually, an estimated 2.86 million cases [...]". Is this globally or only in Africa? • Is it unclear how many Cholera campaigns Nsanje has received. In line 85 it says the country has implemented several campaigns, but two paragraphs below (lines 96-103) you described only one campaign in that District in 2015. Although this number does not affect the results, it helps to understand the context. Methods:  • It is a bit confusing that you mentioned two districts as part of the CSIMA (Nsanje and Chikwawa), but the rest of the manuscript only refers to Nsanje (e.g., abstract and discussion). Please clarify if patients from both districts were included in the study. And if only patients from 
---

	Nsanje were included, please make this explicit. It is confusing that you talk a lot about Chikwawa but it is not part of the study.  • Line 115: the “n=18” and “n=22” in the parentheses of each district should be in the results section, not in methods. In this section the readers do not know the number of patients that you recruited. • Line 116. Is the CSIMA program over or it continues but the period of the study ends in March 2020? • Lines 134-139: when was the inclusion criteria expended? What is the end of the period for the study, 2017 or 2020? Please, clarify it. • Line 155: when do you start counting the days for the interview? From the date of diagnosis or from the date when the patient became symptomatic? • Line 167: what is the minimum age for a patient to be able to answer the interviews by himself/herself instead of being answered by a guardian? • Why patients under 12 months old were excluded? • Lines 178: children >5 years of age may participate in the labor market, as the authors mention, but highly probable they will not be making a full minimum salary. Please add a reference for this assumption or describe how this assumption will affect the results. • Do the productivity losses calculated for the caregivers include those who do not participate in the labor market (e.g., caregivers dedicated to unpaid housework)? • Please add the source of the exchange rate used in the calculations. • Were the costs from different years adjusted by inflation to be presented in a common year? If so, please make it explicit and include the source of inflation. Results:  • Were the 40 patients include in the study only from Nsanje? • Please explain why the direct medical costs for drugs and consumables were zero. Is it because those expenses are covered by the insurance/social security system? I think you mention something related in the discussion, but it would be good to explain that here. • Line 228: Spell out CHAM • Line 248: “(n=12)” seems to be repeated in this sentence. • Line 253: “Half of all patients, and two-thirds of inpatients, [...]”. Please add the exact numbers in parenthesis. • Line 255: how did you define food insecurity for this study? Discussion:  • Line 261: please add the exact value in parenthesis (or in other way) for this phrase “the largest proportion of costs being from direct non-medical costs [...]” • If I understood it correctly, the difference of indirect costs reported in tables 2 and 3 is that in one those are averaged across all patients, while in the other they are averaged only among the patients reporting a wage loss. This difference was confusing, and it would be better if you
--	---

	make this difference clear in the text.  • Line 300: why do Machinga and Zomba districts are relevant? It is the first time they appear in the manuscript and it is not clear why. • Line 321: do patients have to show a document of identity to receive health care? If not, then I do not see why the immigration status of a patient would be relevant. • Line 327: are the \$59.70 in 2019 USD? The values in the next line (329) are in 2018 USD and are not comparable. Could you please present all the numbers in USD of the study in a common year (e.g., 2020)? • Line 340: what is the year of the “\$28 per capita” value (e.g., USD 2019)? • Lines 343-346: you say that the vaccine could be an effective method to reduce the burden of cholera, but this affirmation only can result from a randomized control trial, which is not the design of this study. Table 1:  • The phrasing of the line item “Average number of days prior to seeking medical care (min-max)” is weird. Is this the average number of days since illness onset or diagnosis? Table 2:  • I don't understand why there are indirect costs among the caregivers of patients who received outpatient care if you say in the footnote 2 that only those of inpatient patients reported wage losses. Could you, please, clarify it? Table 3:  • Please clarify the title “% of total”. Is this a % of total costs or total indirect costs? • Footnote 2: what is the “n” for scenario 3.
--	---

VERSION 1 – AUTHOR RESPONSE

Reviewer: 1

Dr. Dzedzom Awalime, The Aurum Institute

Comments to the Author:

The manuscript asserts that Malawi experienced its worst outbreak of cholera in 2015, this should reflect in larger case numbers being sampled, however, only 40 cases were sampled. A focus on cost-saving to households due to the mass vaccination programme undertaken would have been a good approach for this study. Since it could be attributed to the relatively lower number of cases reported.

We thank the reviewer for this feedback. The flooding occurred in January, with cholera being reported in the month following. The vaccination campaign did not begin until March/April, and surveillance did not begin until one year later. We have clarified this limitation in the following section (p. 15, lines 319-324):

“One limitation was the late start of cholera surveillance (April 2016) relative to the cholera outbreak (~February 2015), in response to which vaccination campaign was

conducted (March/April 2015). It is possible that some post vaccination campaign cholera cases may have occurred in the year prior to the start of cholera surveillance, particularly during the November 2015 to April 2016 rainy season; these cases would not have been included in this study.”

We have also added some additional discussion regarding the differences between our study and the other cost of illness study in Malawi, which may help explain the differences in costs (p. 14, lines 305-310):

“The differences may also be attributed to the severity of illness, potentially attenuated by vaccination. Ilboudo et al. assessed costs of illness from patients who had cholera in the ± 1 week prior to and after the OCV vaccination campaign that was underway (which followed the cholera outbreak). Thus, the proximity in timing to the outbreak may have meant that few individuals had any immunity to cholera and may have experienced more severe symptoms, whereas in our population, OCV coverage and immunity was likely higher.”

The data collection process was good and has minimized recall bias. The interval interview process helps for appropriate data to be captured. However, it is not clear if patients that were interviewed at the time of initial outpatient or inpatient visit, were followed-up since this has the potential of missing out on other costs that may occur after this initial phase of care?

The reviewer raises a great question. The initial interview for all cases took place within 7 days of the initial outpatient or inpatient visit. The second interview took place on day 13–15 after the initial encounter. This is described in the methods section (p. 8, lines 156-160):

“A series of a maximum of three interviews were planned for the COI assessment: at day 0–7, day 13–15 (beyond the mean duration of an episode of cholera illness), and day 20-22. To minimize recall bias, the first interview took place as soon as cholera was confirmed either by a test or a clinician, with a maximum of 7 days following the first healthcare facility visit.”

There were only 6 cases that required a second interview, which means that most patients fully recovered by day 7 following the initial recruitment. We have added the following in the results to clarify (p. 10, lines 212-214):

“Most patients (n=34) had fully recovered by the time the first interview occurred (scheduled within 7 days of recruitment day 0); 6 patients had a second interview (at day 13–15) and none required a third interview.”

The interview for costs should be close-ended and must list possible cost drivers. Asking generally on how much money was spent has the tendency for the introduction of unrelated costs or missing out on related costs.

The reviewer raises an important point and we agree that asking generally does not suffice. Our cost of illness survey was very detailed and has been included in this resubmission as a supplementary file. It walks through costs including transportation (cost and method) for the patient to the facility, asks the patient whether anyone accompanied him/her, what the costs/method were for those accompanying, etc. For example, it also goes into detail to ask what was purchased while at the health facility (options: medicine/drugs, diagnostic/lab fee, doctor, nurse, or herbalist fee, other [patient must specify]).

Productivity losses are normally associated with ‘time’ paid jobs or informal sectors. Formal sector are normally entitled to sick days leave hence the individual does not bare productivity losses. The employment status of the sample shows that there were 35 out of 40 patients who work for wages. This does not come out clearly to signify if this cohort experienced productivity losses. Caregiver productivity losses are however captured.

We thank the reviewer for this input. Only 5 patients in our sample reported working and lost wages (not the inverse of 35 patients). The productivity losses are detailed in Tables 2 and 3. In Table 2, productivity losses are calculated based on what was self-reported (averaged the total number of patients for overall costs, n=40), whereas Table 3 presents 3 scenarios where the denominator only includes those who contributed costs.

During the interview, we ask patients the following question: “If you work for a wage, did your (or the patient’s) employer or boss pay for the days you were sick but did not come to work? For example, if the patient has sick leave.” Only one of the patients (out of 5) stated “yes”. To clarify this point, we have added the following to the results section (p. 12, lines 240-243); **bolded** text is added):

“In scenario 1, productivity losses (lost wages) due to being sick for the duration of illness were \$10.60 on average (n=5) for patients who reported wage losses, **though one patient reported receiving paid sick leave for this time (\$10.89 USD), which would reduce the average to \$10.53 (n=4).**”

Though generally, children support the economic activity of parents/caregivers, this is usually done in the context of 'giving a hand' to support the family business. These children do not earn wages and the consideration of this as economic activity has encountered some criticisms especially with child rights advocates. In addition, these kinds of 'hand giving' are usually not contract time-bound. More than 50% (27) of the sample incurred no transportation cost which is usually an important cost driver.

We appreciate the reviewer’s concern and agree that there are ethical issues in counting children. We do not support such practices and have included the scenario 3 sensitivity analysis out of recognition that children may be asked to assist their parents/family. While this may not be paid for in cash, it is still an important cost to society to consider, especially if the child (over the age of 5) could be attending school or other similar activities where an opportunity cost should be considered. For this reason, we do not count children <5 years of age as the opportunity cost is arguably less tangible.

The conclusions of this study are issues widely known. Though the context of Malawi is being presented in this study, the small sample size makes it challenging to use the results from the study as the generalization of the Malawian situation, which can be totally different from what this study has postulated.

We appreciate the reviewer’s concern and agree that this is a limitation of the study. Despite the limitations of the small sample size, however, we feel that the data collection was robust, and the study results are also critically important to report. Cholera cost of illness studies that collect direct evidence from sub-Saharan Africa are very limited. In our discussion, we discuss another study that took place in northern Malawi as a point of comparison. Clearly, there is variation across the limited number of studies in Malawi, which is in part explained by the limited sample size, but likely also a function of the healthcare system and healthcare seeking behavior.

Reviewer: 2

Dr. Debraj Mukhopadhyay, Delhi Pharmaceutical Sciences and Research University

Comments to the Author:

Fair enough to write this.

We thank the reviewer for the positive feedback.

Reviewer: 3

Dr. Nelly Mejia, Centers for Disease Control and Prevention

Comments to the Author:

This manuscript is relevant to understand the economic burden that patients experience during a case of cholera in Malawi. I think the study is well done, although may not be replicable because the volume of cholera cases depend on many external variables. My main comment is that the authors should clarify some parts of the methods and results to make the study more useful.

We thank the reviewer for the positive feedback. We are uncertain which sections the reviewer suggests clarifying; however, we hope that the additions made to the manuscript to the results section make the study more useful.

Results section (p. 10, lines 206-207; **bolded** text added):

“On the day of their health facility visit, all 40 patients also consented for COI interview.”

Results section (p. 10, lines 212-214; **bolded** text added):

“Most patients (n=34) had fully recovered by the time the first interview occurred (scheduled within 7 days of recruitment day 0); 6 patients had a second interview (at day 13–15) and none required a third interview.”

Results section (p. 11, lines 240-243; **bolded** text added):

“In scenario 1, productivity losses (lost wages) due to being sick for the duration of illness were \$10.60 on average (n=5) for patients who reported wage losses, though one patient reported receiving paid sick leave for this time (\$10.89 USD), which would reduce the average to \$10.53 (n=4).”

Discussion section (p. 14, lines 305-310; **bolded** text added):

“The differences may also be attributed to the severity of illness, potentially attenuated by vaccination. Ilboudo et al. assessed costs of illness from patients who had cholera in the ±1 week prior to and after the OCV vaccination campaign that was underway (which followed the cholera outbreak). Thus, the proximity in timing to the outbreak may have meant that few individuals had any immunity to cholera and may have experienced more severe symptoms, whereas in our population, OCV coverage and immunity was likely higher.

Cost-of-illness of cholera in rural southern Malawi from the household Perspective

The authors explore an important topic: the costs of a cholera case from the patient/household perspective in a low-income country. Most studies about cost of illness for different diseases focus on the costs for the provider instead of the patient. The manuscript is well written, and the findings provide a good idea of how burdensome a case of cholera can be for a household in Malawi, even if the direct

medical costs are free (in most cases). However, some parts of the methods are unclear and it would be good if the authors add some clarifications to be sure their methods are consistent and are comparing apples to apples.

We thank the feedback and hope that we have addressed the comments below.

Specific comments below:

Abstract:

- Please clarify in what year are the values provided in dollars (line 38). Is it 2021 USD?

Based on other parts of the reviewers feedback, we have gone through and inflated and then converted all costs to 2020 USD (using the March 31, 2020 USD to Malawian kwacha price since that was the last date of the study). This has been clarified in the abstract (p. 2, line 37). The exact inflation/conversion is also now described in the Methods section (p. 10, lines 184–185):

“Costs were collected in Malawian kwacha (MWK), inflated to 2020 MWK using World Bank inflation data, and converted to 2020 US\$ (736.5803 MWK=\$1 US on March 31, 2020).”

Introduction:

- Line 63: “Annually, an estimated 2.86 million cases [...]”. Is this globally or only in Africa?

This figure refers to cholera globally in cholera-endemic countries. We have clarified in the Introduction (p. 4, line 58–59; **bold** text added):

“Annually, an estimated 2.86 million cases of cholera occur **in cholera endemic countries** that result in 95,000 deaths.”

- Is it unclear how many Cholera campaigns Nsanje has received. In line 85 it says the country has implemented several campaigns, but two paragraphs below (lines 96-103) you described only one campaign in that District in 2015. Although this number does not affect the results, it helps to understand the context.

We thank the reviewer for this comment. The current cost of illness study follows the first oral cholera campaign (OCV) in Malawi. The “several” described in the earlier paragraph refer to campaigns that have taken place since the first. We appreciate how this can be confusing and have moved the following text from the introduction to the discussion (p. 17 lines 363–366; **bolded** text was edited):

“**Since this study was conducted**, in recognition of the high incidence of cholera and the need for additional tools to address the burden, in 2017, the government of Malawi integrated OCVs into its national cholera control plan. and there have been several OCV campaigns implemented since.”

Methods:

- It is a bit confusing that you mentioned two districts as part of the CSIMA (Nsanje and Chikwawa), but the rest of the manuscript only refers to Nsanje (e.g., abstract and discussion). Please clarify if patients from both districts were included in the study. And if only patients from Nsanje were included, please make this explicit. It is confusing that you talk a lot about Chikwawa but it is not part of the study.

We can understand this confusion. In the initial study design, the intent was to also conduct a vaccine effectiveness and vaccine cost effectiveness study (currently underway in a separate analysis). Thus, it was important for a relatively large number of residents to be vaccinated. Due to limited resources, the vaccines were targeted toward Nsanje residents that comprise the majority of those affected the floods. Given the complexities of implementing a surveillance system in this region of Malawi, and limited personnel/funding, we also decided

to focus our efforts on COI data collection in Nsanje where the majority of the vaccines were used. We feel that including the detail about Chikwawa surveillance is an important point and significant portion of the overall surveillance effort though Chikwawa residents were not included for COI.

In the Methods section we currently state the following (p. 7, lines 135-137):

“While cholera vaccines were administered to residents in both Nsanje and Chikwawa, the majority of doses were administered to residents of Nsanje. Thus, only patients recruited from Nsanje health centers were invited for the COI portion of the study.”

We have added the following in hopes that this further clarifies that Chikwawa was not included (p. 15, lines 324-325; **bold** text new):

“The study was also not able to collect COI data from Chikwawa residents and infants <12 months, which may have provided different cost estimates.”

- Line 115: the “n=18” and “n=22” in the parentheses of each district should be in the results section, not in methods. In this section the readers do not know the number of patients that you recruited.

We thank the reviewer for this correction. The numbers refer to the number of health facilities included in the surveillance program (as opposed to the number of patients, which we agree should be in the results section). We have clarified in the Methods section as follows (p. 6, lines 107-109; **bold** is revised text):

“As part of the Cholera Surveillance in Malawi (CSIMA) program, cholera surveillance began across all **22** Nsanje (population >292,000) and **18** Chikwawa (population >533,000) healthcare facilities from April 5, 2016, to March 31, 2020.”

- Line 116. Is the CSIMA program over or it continues but the period of the study ends in March 2020?

We have clarified the language in this sentence as follows so that “began” is replaced by “was conducted.” The program has ended (though the surveillance efforts continue without IVI’s direct involvement) (p. 6, lines 106-108; **bold** is revised text):

“As part of the Cholera Surveillance in Malawi (CSIMA) program, cholera surveillance **was conducted** across 22 Nsanje (population >292,000) and 18 Chikwawa (population >533,000) healthcare facilities from April 5, 2016, to March 31, 2020.”

- Lines 134-139: when was the inclusion criteria expended? What is the end of the period for the study, 2017 or 2020? Please, clarify it.

We thank the reviewer for pointing out this section. We have clarified the language in this paragraph (p. 7, lines 128-133; **bold** text added):

“Between April 6, 2016, and December 12, 2017, culture-confirmed and RDT-positive cholera cases were recruited for the overall study, as well as the COI survey component. However, due to low recruitment numbers of RDT- and/or culture-confirmed cholera cases, in part due to limited availability of RDT kits and stool collection logistical challenges, the inclusion criteria were expanded **from December 12, 2017, through March 31, 2020**, to include clinically-diagnosed cholera cases in an outbreak situation to avoid missing potential cases.”

- Line 155: when do you start counting the days for the interview? From the date of diagnosis or from the date when the patient became symptomatic?

We thank the reviewer for the question. We count the days for the interview from day 0, the day of the health facility visit when we (the study team/health facility) first encounters the patient. The first interview includes questions about any health facility visits prior to day 0, and asks the patient when symptoms first began. We have clarified as follows (p. 8, lines 148-150; bold text added):

"A series of a maximum of three interviews were planned for the COI assessment, **with day 0 (i.e., day of diagnosis) being the day of the patient's health facility visit**: at day 0–7, day 13–15 (beyond the mean duration of an episode of cholera illness), and day 20–22."

- Line 167: what is the minimum age for a patient to be able to answer the interviews by himself/herself instead of being answered by a guardian?

For patients who were <12 years of age, consent was given by the parent/guardian, and in these cases, the patient/caregiver was the one who provided the interview responses. For patients aged 12 through 17 years, ascent was given by both the parents and child. Adults 18 years and older provided consent directly.

- Why patients under 12 months old were excluded?

Patients <12 months of age were excluded, since they were not eligible to receive the vaccine. While the COI study was meant to collect costs of illness associated with cholera (regardless of vaccination status), one of the sub-studies of the overall project was to evaluate vaccine cost effectiveness, which would require a comparison of similar vaccinated and unvaccinated cohorts. We do not include this level of detail in the manuscript as it may detract from the main point regarding costs. However, we have added a sentence to the limitations section (p. 15, lines 324-325; **bold** text added):

"The study was also not able to collect COI data from Chikwawa residents **and infants <12 months**, which may have provided different cost estimates."

We have clarified in the Methods section as well (p. 6, lines 115-118; **bold** text added):

"Patients who met the inclusion criteria (**who were eligible to receive OCV based being** at least 12 months of age at the start of the vaccination campaign, and presented to an Nsanje health facility with acute watery diarrhea indicative of cholera as defined by the WHO13) were invited to participate in the diarrheal surveillance study."

- Lines 178: children >5 years of age may participate in the labor market, as the authors mention, but highly probable they will not be making a full minimum salary. Please add a reference for this assumption or describe how this assumption will affect the results.

We appreciate the reviewer's concern that including a full minimum salary may skew the actual lost productivity costs. We run 3 scenarios, the first of which only includes self-reported wages. In some instances, a patient reported making wages, but reported that illness had zero impact on their wages; therefore, such patients do not incur productivity losses.

Our intent for including the sensitivity analysis with all persons >5 years of age was to ensure that contributions to the household are valued from a societal perspective. Particularly for children of this age, while they may not legally be allowed to work, the opportunity cost of missing school to help family is considerable and we felt that the fairest market value was to at least use the minimum wage (\$1.83USD/day).

We have added the following to the Results (p. 12, lines 247-248):

“In this scenario analysis, there were 2 patients aged 5–14 years (\$14.16 on average) and 5 patients aged ≥60 years (\$8.59 on average).”

We have also added the following to the Discussion to describe how our assumptions affect the results (p. 13, lines 279-285; **bold** text added):

“In our scenario analysis where we applied the minimum wage to all patients and caregivers who theoretically could be contributing to economic activities (i.e., household work), indirect productivity losses were as high as \$9.78 per patient and \$3.94 per caregiver. **This latter scenario may overestimate the number of individuals who experience productivity losses, though the average productivity losses (\$13.72) were lower than in the self-reported wage scenario (\$16.96).**”

- Do the productivity losses calculated for the caregivers include those who do not participate in the labor market (e.g., caregivers dedicated to unpaid housework)?

We thank the reviewer for the question. We do not include productivity losses for caregivers who report \$0 income lost.

- Please add the source of the exchange rate used in the calculations.

This reference for the historical exchange rate on March 31, 2020, has been added (reference #21).

- Were the costs from different years adjusted by inflation to be presented in a common year? If so, please make it explicit and include the source of inflation.

We thank the reviewer for this question. The reviewer makes an important point. This was our oversight and inflation was not applied. Therefore, we have gone back and applied inflation to each individual-level observation in Malawian kwacha (inflate to 2020 kwacha). After inflating, we then convert to USD using the middle exchange rate provided the Reserve Bank of Malawi for March 31, 2020 (the last day of our study). We have clarified this in the manuscript as follows, with the appropriate references added to the manuscript (references 21 and 22) (p. 9, lines 184-185; **bold** text added/edited):

“Costs were collected in Malawian kwacha (MWK), **inflated to 2020 MWK using World Bank data**, and converted to 2020 US\$ (**736.5803 MWK=\$1 US on March 31, 2020**).”

Results:

- Were the 40 patients include in the study only from Nsanje?

Yes, all COI cases were only recruited from Nsanje. We have added this to the results section to clarify (p. 10, lines 204-205; **bold** text added):

“During our surveillance period, 2,201 individuals met the diarrheal illness criteria for stool collection and testing for cholera **in Nsanje**.”

- Please explain why the direct medical costs for drugs and consumables were zero. Is it because those expenses are covered by the insurance/social security system? I think you mention something related in the discussion, but it would be good to explain that here.

The reviewer is correct that this is currently mentioned in the manuscript in the following sections:

Abstract, conclusion (p. 2, lines 41-45):

“Conclusion: For the majority of Malawians who struggle to subsist on less than \$2 a day, the COI of cholera represents a significant cost burden to families. While cholera treatment is provided for free in government-run health centers, additional investments in cholera control and prevention at the community level and financial support beyond direct medical costs may be necessary to alleviate the economic burden of cholera on households in southern Malawi.”

We have moved text up from the latter part of the discussion to the second paragraph as follows (p. 13 lines, 272-274; **bold** text added/moved):

“Overall, only two cholera patients had any DMCs; one was seen in the outpatient setting (875 MWK, or \$1.19) and the other was admitted to a private CHAM facility (8,229 MWK, or \$11.17). **This finding was expected, as cholera treatment is provided free of charge in Malawi.**”

- Line 228: Spell out CHAM

Thank you for catching this abbreviation. We have spelled it out on first reference (“Christian Health Association of Malawi”).

- Line 248: “(n=12)” seems to be repeated in this sentence.

We thank the reviewer for catching this error. We have deleted the second instance of this.

- Line 253: “Half of all patients, and two-thirds of inpatients, [...]”. Please add the exact numbers in parenthesis.

We have added this to p. 12, lines 252-254; **bold** text added):

“Half of all patients (**n=20**), and two-thirds of inpatients (**n=6**), reported having to borrow money from someone to pay for health care, food, or transportation as a result of having cholera (Table 4), but most did not have to sell personal belongings to pay for care.”

- Line 255: how did you define food insecurity for this study?

We thank the reviewer for this question. Table 4 lists the questions that were asked of COI patients. For food insecurity specifically, we ask them how frequently they went without food in the past year. The possible responses are listed in the rows of that section of the table. (The patient also had the option of giving an “other” response.)

Discussion:

- Line 261: please add the exact value in parenthesis (or in other way) for this phrase “the largest proportion of costs being from direct non-medical costs [...]”

We have added this to the sentence (which references the “% of total” column in Table 2).

- If I understood it correctly, the difference of indirect costs reported in tables 2 and 3 is that in

one those are averaged across all patients, while in the other they are averaged only among the patients reporting a wage loss. This difference was confusing, and it would be better if you make this difference clear in the text.

We can appreciate the confusion and have clarified in the text as follows in the Results section (p. 11, lines 219-221; **bold** text added):

“The direct and indirect cost of cholera per patient was \$14.34 **when averaged across all patients**, with the largest proportion of costs (57.9%) being from direct non-medical costs (food, water, and transportation) (Table 2).”

We have also added a footnote for scenario 1 and revised the column header as “**Among those with** self-reported wages only”.

- Line 300: why do Machinga and Zomba districts are relevant? It is the first time they appear in the manuscript and it is not clear why.

We can understand how this can be confusing. In the introduction, we state that only 1 prior COI study was conducted in Malawi, which was done in the Lake Chilwa area. Lake Chilwa resides in the districts of Machinga and Zomba. To clarify this to the reader upfront, we have added this to the Introduction so that it does not seem to be abruptly introduced in the Discussion (p. 5, lines 100-101; **bold** text added):

“Only one study to date has reported on the cost-of-illness from cholera in **southeastern Malawi (Lake Chilwa in the districts of Machinga and Zomba)**.”

- Line 321: do patients have to show a document of identity to receive health care? If not, then I do not see why the immigration status of a patient would be relevant.

We appreciate the reviewer’s comment. Our understanding is that showing documentation is not required in Malawi. However, even the perception of deportation or perception that health care could be denied for any reason can be enough to deter one from seeking care. We have added a reference to the following article from Legido-Quigley, et al. in *BMJ* (<https://www.bmj.com/content/366/bmj.l4160>) and edited the text to illustrate other reasons why a Mozambican might not seek care (p. 16, lines 337-338; **bold** text added):

“...migrants may be less likely to seek care **due to several reasons, including language barriers, discrimination due to cultural differences, and** fear of deportation.”

- Line 327: are the \$59.70 in 2019 USD? The values in the next line (329) are in 2018 USD and are not comparable. Could you please present all the numbers in USD of the study in a common year (e.g., 2020)?

We thank the reviewer for pointing this out. We have added the 2020 conversion after inflating for comparability. For example, we first convert \$59.70 in US\$ 2016 to Malawian kwacha 2016. We then inflate this figure to 2020, and then convert back to USD using March 31, 2020.

- Line 340: what is the year of the “\$28 per capita” value (e.g., USD 2019)?

We have reviewed the reference again (<https://www.resakss.org/sites/default/files/Malawi%20MH%202011%20Malawi%20Health%20Sector%20Strategic%20Plan%202011%20-%202016.pdf>). There has been an update to the document since we last reviewed it. Therefore, we have edited the sentence taking out the reference to \$28 per capita.

We have replaced the sentence with the following and revised reference (https://extranet.who.int/countryplanningcycles/sites/default/files/planning_cycle_repository/malawi/health_sector_strategic_plan_ii_030417_smt_dps.pdf), which conveys our key takeaway that spending is inadequate and unsustainable in Malawi to meet the needs of its population (p. 16, lines 355-357):

“Health care financing in Malawi is also unpredictable and unsustainable given the country’s reliance on development partners’ contributions that account for nearly two-thirds of total health expenditure.”

- Lines 343-346: you say that the vaccine could be an effective method to reduce the burden of cholera, but this affirmation only can result from a randomized control trial, which is not the design of this study.

We thank the reviewer for this comment. There are clinical trials demonstrating the vaccine effectiveness (and by extension reduced burden) of the oral cholera vaccine. We agree with the reviewer though that our study was not designed to be a vaccine effectiveness study, and there are several limitations to the setting that make it very difficult to assess this properly. Therefore, we have softened the language. The main point we wanted to convey here is that it is possible that the vaccine averted cholera illness, but of course we have no definitive way of ascertaining this in the current study. Text has been edited as follows (p. 17, lines 360-363; **bold** text added):

“**While the study was not designed to assess vaccine effectiveness**, our surveillance efforts may also be a positive indication that the cholera vaccine **may be associated with a reduction in the** burden of cholera in the middle-term, as most of our cases were unvaccinated (approximately half of the Nsanje population received the cholera vaccine).”

Table 1:

- The phrasing of the line item “Average number of days prior to seeking medical care (min-max)” is weird. Is this the average number of days since illness onset or diagnosis?

The reviewer is correct. This is the average number of symptomatic days a patient waited prior to seeking care. We have edited this as “Average number of **symptomatic** days prior to seeking medical care (min–max).”

Table 2:

- I don’t understand why there are indirect costs among the caregivers of patients who received outpatient care if you say in the footnote 2 that only those of inpatient patients reported wage losses. Could you, please, clarify it?

We thank the reviewer for pointing this out and acknowledge that it is confusing and misleading. We have corrected the footnote; a subset of inpatient *and* outpatient patients had caregivers who had productivity losses. We have clarified the footnote as follows:

“Productivity losses are averaged over the total number of patients in each category (not averaged over the total number of caregivers); 6 outpatients and 2 inpatients had caregivers who only two caregivers (of two inpatient patients) had any reported wage losses, but wages are averaged over the total number of outpatients (n=31) and inpatients (n=9)”

Table 3:

- Please clarify the title “% of total”. Is this a % of total costs or total indirect costs?

We have clarified this as “% of total indirect costs.”

- Footnote 2: what is the “n” for scenario 3.

We have clarified this portion of the table. We have added “n’s” for each scenario at the top of the columns so that it is more clear to the reader. The “n” for caregivers in scenario 2 and 3 are the same.

VERSION 2 – REVIEW

REVIEWER	Awalime, Dziedzom The Aurum Institute, International
REVIEW RETURNED	24-Feb-2022

GENERAL COMMENTS	Malawi had a cholera vaccination drive in 2015 and cholera surveillance was initiated from 2016 to 2020. This study sort to capture the cost of illness of cholera post a vaccination drive hence very limited number of cases were reported and among those reported cases majority were unvaccinated. Though per definition one case of cholera is an outbreak, for the study to adequately capture the impact of the COI on the society required higher incidence. However, this revised version of the work adequately captures the main limitations of the study clearly. Limitations of small sample size, recall bias, and missing data are well addressed. Scenarios of productivity losses were well documented to reflect the likely biases. Though the study would have been enriched by the capture of facility costs, the study's focus was primarily on household costs. Important areas of a cost of illness study such as the duration of illness were captured. The methodology applied was adequate to capture the objectives of the study.
---

REVIEWER	Mejia, Nelly Centers for Disease Control and Prevention
REVIEW RETURNED	22-Feb-2022

GENERAL COMMENTS	The authors have addressed almost of my comments and have improved the clarity and quality of the manuscript. However, there are three comments that need further work. Two of them were answered in the response document that they sent me, but they did not add any clarifying text to the manuscript; and one question (about food insecurity) was not addressed as the term remains undefined. I attached the three pending comments with my follow-up answers I would like to encourage the authors to address these three comments. I have no further feedback besides this one and I do not think the manuscript needs another round of revision. Reviewer: 3 Dr. Nelly Mejia, Centers for Disease Control and Prevention  • Line 167: what is the minimum age for a patient to be able to answer the interviews by himself/herself instead of being answered by a guardian? For patients who were <12 years of age, consent was given by the parent/guardian, and in these cases, the patient/caregiver was the
--

	one who provided the interview responses. For patients aged 12 through 17 years, ascent was given by both the parents and child. Adults 18 years and older provided consent directly. Thank you for answering my concern. Could you add this explanation to the manuscript? This would help the reader to understand any recall bias depending on the survey respondent.  • Do the productivity losses calculated for the caregivers include those who do not participate in the labor market (e.g., caregivers dedicated to unpaid housework)? We thank the reviewer for the question. We do not include productivity losses for caregivers who report \$0 income lost. Thanks for your answer. Could you please mention it in the text?  • Line 255: how did you define food insecurity for this study? We thank the reviewer for this question. Table 4 lists the questions that were asked of COI patients. For food insecurity specifically, we ask them how frequently they went without food in the past year. The possible responses are listed in the rows of that section of the table. (The patient also had the option of giving an “other” response.) The definition of food insecurity is still unclear. The text reports 35 people in food insecurity, but I could not find that number in Table 4. When I ask it to define it is because I would like to know what categories from the table are included, so I can add them up and come up with the n=35.
--	--

VERSION 2 – AUTHOR RESPONSE

Reviewer: 3

Dr. Nelly Mejia, Centers for Disease Control and Prevention

- Line 167: what is the minimum age for a patient to be able to answer the interviews by himself/herself instead of being answered by a guardian?

For patients who were <12 years of age, consent was given by the parent/guardian, and in these cases, the patient/caregiver was the one who provided the interview responses. For patients aged 12 through 17 years, ascent was given by both the parents and child. Adults 18 years and older provided consent directly.

Thank you for answering my concern. Could you add this explanation to the manuscript? This would help the reader to understand any recall bias depending on the survey respondent.

We thank the reviewer for this feedback. We agree it's important and have added it to the manuscript on p. 10, lines 194–197:

“For patients aged <12 years, consent was provided by the parent/guardian, and in these cases, the patient/caregiver provided the interview responses. For patients aged 12–17 years, ascent was provided by both the parent/guardian and child. Adults aged ≥18 years provided consent directly.”

- Do the productivity losses calculated for the caregivers include those who do not participate in

the labor market (e.g., caregivers dedicated to unpaid housework)?

We thank the reviewer for the question. We do not include productivity losses for caregivers who report \$0 income lost.

Thanks for your answer. Could you please mention it in the text?

We thank the reviewer for this feedback. We have added this to the text, and attempted to further clarify how wage calculations were performed in the 3 scenarios (as it may have been confusing that the minimum wage was being applied to all scenarios. We have edited/added the text on p. 9-9, lines 176-184 (bold and underlined text edited/added):

“In all scenarios, if patients and/or caregivers reported wage losses, these were calculated based on self-reported daily lost productivity (half or full days of work lost due to taking care of the ill patient) and daily wages. **For patients and caregivers who did not report wage losses (i.e., scenarios 2 and 3)**, we applied the minimum wage of 1,346.16 Malawian kwacha (MWK) per day (\$1.83 USD/day as of January 1, 2020) to account for the monetary value of productivity losses. Productivity losses were not calculated for caregivers who indicated that they did not cut back on their usual activities had they not been caring for the patient. **They were also not calculated for caregivers who reported no income lost (e.g., a caregiver who is dedicated to unpaid housework).**”

• Line 255: how did you define food insecurity for this study?

We thank the reviewer for this question. Table 4 lists the questions that were asked of COI patients. For food insecurity specifically, we ask them how frequently they went without food in the past year. The possible responses are listed in the rows of that section of the table. (The patient also had the option of giving an “other” response.)

The definition of food insecurity is still unclear. The text reports 35 people in food insecurity, but I could not find that number in Table 4. When I ask it to define it is because I would like to know what categories from the table are included, so I can add them up and come up with the n=35.

We thank the reviewer for this careful review and apologize for this oversight. Our initial calculation was intended take the total number of participants (n=40) and subtract out the 5 who specified that they never experienced going without food in the past year. We now realize that this overlooks the missing/unknown category (n=7). We have made this correction and clarified in the text how we define food insecurity. We feel that it is important to take a more liberal approach to including anyone who experiences any food insecurity, and thus, being able to recall going without food even once in the past 3 months was counted. The text is revised as follows (p. 12-13, lines 261-263):

“Experiencing at least some food insecurity in the past year (defined as going without food at least once in the past 3 months) was an issue for a large proportion of patients (n=28).